# Peek-At-You: An Awareness, Navigation, and View Sharing System for Remote Collaborative Content Creation

Matthew K Miller*

University of Saskatchewan

Frederik Brudy†

Autodesk Research

Tovi Grossman‡

University of Toronto

George W. Fitzmaurice†

Autodesk Research

Fraser Anderson†

Autodesk Research

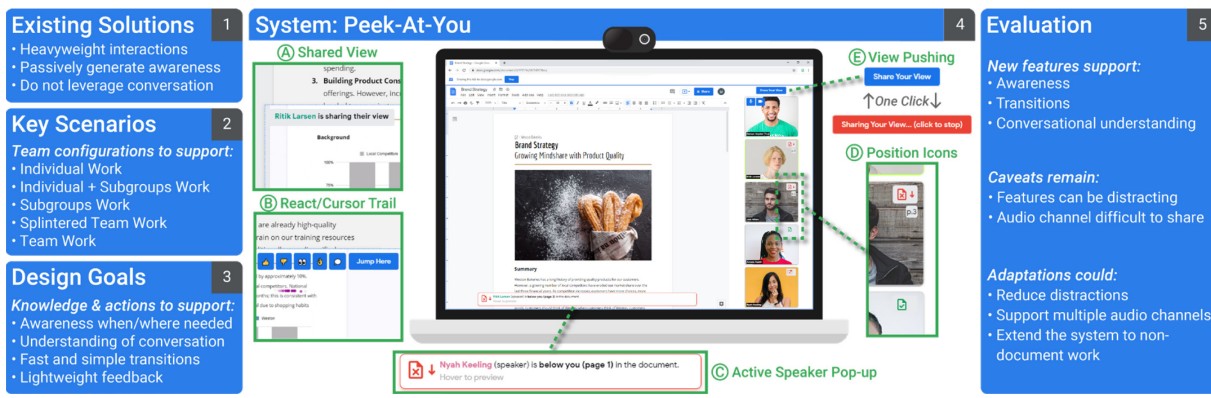

Figure 1: Overview of our research process

## ABSTRACT

Remote work plays a critical and growing role in modern workplaces. A particular challenge for remote workers is mixed-focus collaboration, which involves frequent switching between individual and group tasks while maintaining awareness of others' activities. Mixed focus collaboration is important in content creation as it can benefit from the greater perspective, larger skill set, and reduced bias of a group, but this work is difficult to do remotely because existing systems only provide information about collaborators passively or through cumbersome interactions. In this paper, we present Peek-at-You, a system of collaborative features leveraging integration between collaboration and communication software, including conversational position indicators, speaker's view peeking, and view pushing. Our evaluation shows these features help support awareness, understanding, and working state transitions. Finally, we discuss adapting the features to manage distractions and support various work artifacts.

**Keywords**: Groupwork, Remote Collaboration, Content Creation

**Index Terms**: Human-centered computing—Collaborative and social computing—Collaborative and social computing theory, concepts and paradigms—Computer supported cooperative work

## 1 INTRODUCTION

Distributed teams require successful technology-based collaboration to functioning effectively [1]. Collaborative work, in which actions are "influenced by the presence of, knowledge of, or the activities of another person" [2, p. 145] is necessary for distributed teams and can be conducted asynchronously or synchronously. We focus on synchronous collaboration, which has become rarer among remote workers due to barriers not solved by existing systems [3]. Recent shifts to remote work due to COVID—19 are associated with a decrease in synchronous communication [4]. Ellis et al.'s time-space matrix refers to this type of collaboration as "same time, different place" [5].

Synchronous collaboration comes in various forms, such as parallel work on separate tasks or collective work on one task. However, for remote teams, the more complex mixed-focus collaboration presents challenges [6]. This type of collaboration involves moving back and forth between individual tasks and shared work with other group members while maintaining awareness of their whereabouts and activities [7]. Quick and fluid transitions between individual and shared work are key to successful mixed-focus collaboration [8]. Poor support for transitions can add significant friction to collaboration.

Collaborative content creation, an instance of mixed-focus collaboration, may be undertaken collaboratively for many reasons, e.g., distributing tasks, leveraging different expertise, avoiding bias, and gaining multiple perspectives [9]. It rates highly on the *shared task* and *shared environment* dimensions of collaboration [5] as people aim to create a cohesive final artifact. In contrast to *collaborative writing*, which has received attention in early groupware systems (e.g., [5], [10], [11]), we focus on the broader term "*content creation*" as formatting and graphical capabilities in modern groupware go beyond simple text and we design for steps other than writing, such as researching and decision making.

Ishii et al. described two types of collaborative spaces that tools can create: *shared workspaces* (spaces that allow sharing information, pointing to specific items, marking, etc.) and *interpersonal space* (spaces that allow verbal and nonverbal communication, eye contact, etc.) [12]. Synchronous remote collaboration requires multiple tools, such as audio and video calls for interpersonal space and real-time groupware for shared workspaces [13]. However, combining these tools can result in a

*email: matthew.miller@usask.ca

‡email: tovi@dgp.toronto.edu

†email: {firstname.lastname}@autodesk.com

disparate and unoptimized set of tools, such as conflicting signals and inaccurate perceptions. For example, screensharing allows everyone to see the same thing but does not allow everyone to work on it and transitioning between different user's sharing is tedious. Further, awareness widgets (e.g., Mural's presence icons or mini-map; Figure 2 bottom) are passive and may not provide accurate awareness of ongoing conversation and transitioning from individual to subgroup work requires searching a list of users or decoding colors to find out what part of an artifact others are working on. Lastly, the combination of signals from shared workspace and interpersonal space may lead to conflicting signals or inaccurate perceptions. For example, in video chat a collaborator's face becomes visible when they speak, giving the impression that they share a perspective on the workspace, even if they are seeing different parts of an artifact.

To address these challenges, we propose *Peek-at-You*, a system that integrates communication and collaborative software with elements that respond to users' activities and conversations. *Peek-At-You* includes Conversation-Based Location Indicators that overlay icons on users' video feeds to show which part of an artifact they are looking at (Figure 1D); a popup over the shared artifact to indicate the current speaker's position (Figure 1C); Speaker's View Peeking which allows users to preview the active speaker's view without leaving their current location (Figure 1A); and View Pushing (Figure 1 E) which streams a user's view as an overlay in the collaborative application, enabling quick transitions between individual and shared work. These features illustrate the potential of integrated systems for remote mixed-focus content creation.

This work makes three main contributions: 1) a series of formative observations using existing tools that contribute to our system's development. 2) *Peek-At-You*, a system and prototype implementation extending Google Docs. 3) findings form our evaluation, that show how integrated systems can foster the awareness, understanding, and transitions critical to enacting mixed-focus collaboration. We further discuss design iterations to minimize distractions and adapt the system to a variety of artifacts. By expanding the understanding and system support for mixed-focus content creation, our work advances the ability of systems to support synchronous collaboration for remote workers.

## 2 RELATED WORK

Our work builds upon three areas: 1) systems for synchronous remote collaboration; 2) tools for supporting fundamentals of remote collaboration, and 3) specific characteristics and requirements of mixed-focus collaboration.

### 2.1 Synchronous remote collaboration

Distributed teams "work together on a mutual goal or work assignment, interact from different locations, and therefore communicate and cooperate by means of information and communication technology" [14, pp. 459–460]. Remote collaboration plays a growing role [15], [16] in many types of work [17]–[20].

Video-mediated communication (i.e., VMC or video chat) is a useful tool for supporting remote teams, work and learning [20]–[22]. VMC apps often allow text chat and screensharing [23], but are not aware of interactions in other collaborative apps.

Another important tool for remote work is Real-time groupware, which lets "distance-separated people work on a shared task in real time" [13, p. 66] (e.g., editing documents, slides, or interface prototypes). Traditionally, real-time groupware

has not integrated communication (e.g., [24]–[26]), even relying on speaking over room dividers for studies (e.g., [27], [28]). Many modern apps also act as "silos" with little integration (e.g., Slack and Zoom focus on communication while Figma and Microsoft Word focus on content). Increasingly, VMC is being integrated with groupware (e.g., video calls in Google Docs [29] or whiteboards [30] and third-party apps [31] in Microsoft Teams calls), but conversational data such as the current speaker is not used to support collaboration.

Some research projects have sought to integrate the shared workspace with the interpersonal space: Ishii et al. did so for pairs by overlaying drawing atop a remote partner's video feed [12] and Grønbæk et al. did so for groups by allowing both spaces to become semi-transparent and overlay each other [32]. However, these systems have limitations including number of users, manually managing positions and opacities, and interfaces that differ significantly from single-user apps. Peek-At-You uses a logical integration: it surfaces workspace data in the interpersonal space and vice-versa, but does not overlay the two; this builds on the familiarity and scalability of traditional groupware and VMC.

Collaborative content creation tasks involve working with others to generate content; writing is a popularly studied example (e.g., [33]–[37]), but other examples include presentations and interface designs. Research suggests remote content generation is associated with less communication, more focus on organization of work, and less focus on feedback or content [37]. One particular aspect that collaborators negotiate is territories, i.e., portions of an artifact that are primarily edited or controlled by one person. These can both explicit or implicit and vary in duration [34]. Collaborators also negotiate transitions between tools (e.g., a document editor, notepad, and LaTeX editor) depending on their current needs [36].

### 2.2 Views, awareness, and gestures

Screensharing is a longstanding paradigm for making apps "collaborative" [38]. Screensharing is asymmetrical: typically, one person shares at a time, decides what is shown, and manipulates the interface. "Remote control" may allow another user to move the cursor, but does not support multiple real-time collaborators [38]. Screensharing provides WYSIWIS ("what you see is what I see") collaboration, but real-time groupware can go beyond this limitation as people use their own instance of the software. Real-time groupware can be WYSIWIS [39] or relaxed-WYSIWIS: viewports, representations, and formatting, and can vary per-user [35], [40]. This increases the Independence of users [41]. While tools that spatially integrate interpersonal space and workspace use screensharing and WYSIWIS views to mix content and VMC [32], [42], Peek-at-You benefits from the independence and shared control via integrated relaxed-WYSIWIS groupware.

In relaxed-WYSIWIS groupware, it can be difficult to maintain workspace awareness—i.e., "the up-to-the-moment understanding of another person's interaction with the shared workspace" [6, p. 417]. Awareness includes multiple elements: who (presence, identity, authorship), what (action, intention, artifact), and where (location, gaze, view, reach) [6]. Gutwin et al. [43] devised several awareness supports, including Radar Views (a scaled down overview of the workspace), Multiple-WYSIWIS Views (scaled down mirrors of others' views), WYSIWID Views (a full-size view of the area around another's cursor), and Teleportals (temporary navigation to someone's viewport). Showing another user's screen or the area around their cursor is helpful, but poorly scales to groups because of limited screen real-estate [7]. More generally, existing awareness supports have significant drawbacks: because they are not aware of who the user is

communicating with, they cannot optimize screen usage or highlight the most relevant information.

Supporting awareness involves a tradeoff with distractions. This can relate to usage of screen real-estate, visual feedback of others' work [7], and collaborators interrupting. For best results, care should be taken before, during, and after interruptions, to ensure that it occurs at an opportune time, the interruption is handled completely, and the original task is resumed easily [44]. There are multiple approaches to this, including immediate interruptions, negotiating when an interruption will occur, or having a mediator or schedule for interruptions [45]. Systems can employ these directly or support people in using them.

Like awareness, gestures and references are also critical elements of collaboration that are difficult to leverage in remote contexts [46]. Gestures allow people to communicate things that are difficult to verbalize, e.g., where an item is located, and occur very frequently during face-to-face collaboration [46], [47]. One common type of gesture, deictic referencing, involves pointing to establish what object a person is referring to as they speak [48]. "Pointing" using a remotely displayed cursor (i.e., a telepointer) is common, but with relaxed-WYSIWIS the content being pointed to may be rendered differently or even outside remote user's viewport [40]. References and gestures also require we-awareness ("the socially recursive inferences that let collaborators know that all are mutually aware of each other's awareness" [49, p. 279]). The first step of gestures is establishing mutual orientation ("that both parties can see the gesture and the target") [50, p. 1378], so systems must allow collaborators to establish a shared view and also be aware of this state. Our system allows people to quickly establish mutual orientation, by pushing a view and seeing the current viewers or by jumping to others' positions and seeing who is in the same area.

### 2.3 Configurations, transitions, and activities in mixed-focus collaboration

Broadly, mixed-focus collaboration involves "individual tasks … and shared work" [7, p. 207]. To be more specific, the Coupling typology characterizes work as Light-weight Interactions, Information Sharing, Coordination, Collaboration, or Cooperation [51]. Mixed-focus collaboration occurs at the more tightly coupled levels, which are rarely done remotely [3]. Another way to characterize group work is subgroupings. Informally, this may include parallel (individual), pair/small-group, and group work [52]; formally, subgrouping can be described in more detail [53]. A third way to characterize groupwork is content focus. For example, one such categorization includes discussion, view engaged, sharing of the same view, same information but different views, same specific problem, same general problem, different problems, and disengaged [54]. These characterizations raise key concepts—coupling, groupings, and shared views—that we use to define important configurations for our system to support.

In mixed-focus collaboration, transitions between working configurations are key to success [8]. Transitions facilitate the three typical phases of collaboration: pre-process, in-process, and post-process [55]. Further, transitions facilitate various activities while in-process (e.g., creating content, presenting results, comparing results, and sharing content) [56]. Several research projects seek to support transitions. For classrooms, one allows teachers to plan and make planned or fluid transitions between individual, small-group, and whole-group phases [57]. For in-person collaboration, shape-changing furniture can aid transitions [58] or an extra shared device can aid moves from individual to group work [59]. For remote work between pairs, continuous screen sharing using a second monitor supports transitions [60].

For other remote work, the TeamWave system uses a room metaphor to ease transitions [61]. The Peek-At-You system supports transitions for fully remote groups, with a design intended for a variety of artifacts.

### 2.4 Support for collaborative content creation

Creating content collaboratively requires both planning (defining the goals for the content, discussing resources of each collaborator, defining the forms of collaboration to occur, and allocating tasks) and production (sketching, composing, and reviewing content an individual and group level) [37]. The collaborators must communicate, coordinate, cooperate, and maintain awareness [51].

Commercial and research systems have worked to advance support for these key elements. Video chat supports conversations and awareness [62], which is important for planning (e.g., discussing how to distribute tasks) and production (e.g., reviewing others' individual work through discussion or speaking about how to compose individual work into a cohesive whole). Relaxed-WYSIWIS groupware allows people to do individual sketching or composition work (by taking on their own views) as well as group composing and reviewing work (because the task space is shared) [13]. Within groupware, awareness tools provide support for monitoring and understanding what others in the group are doing [43]. However, research suggests that existing tools still require remote groups to spend a large amount of time organizing their work, limiting their ability to focus on planning and discussing the content itself [37]. Research testing non-traditional spatial interfaces has shown that combining communication and collaborative tools has the potential to further support for communication, organization, and awareness [12], [32]. To best support collaborative content creation, we consider a non-spatial approach to integrating task and interpersonal space, seeking to support communication, awareness, and group work processes while maintaining the familiar interfaces of productivity and communication tools.

Integrating and sharing data between the task and interpersonal spaces may offer many benefits. First, it could reduce the burden of managing windows [63] and help avoid a sense of impoliteness related to multitasking [64]. Second, since collaborators using VMC spend 5-17% of the time looking at the video feeds [65], [66], an integrated system could place awareness widgets near video feeds to make them more consistently visible. Third, awareness indicators on peoples' video feeds could tie information to easily scannable of video feeds, rather than a row of circles that must be searched and interacted with to locate others' positions. Signals could also be prioritized based on the current speaker. Fourth, integration could enable unique view sharing tools, reducing difficulties with starting and managing shared views [7]. For example, privacy preserving 'push' and 'pull' view sharing, quick transitions to co-editing, highly visible gesture cursors only when needed, and prioritized access to the current speaker's view. Fifth, an integrated approach could support multiple working styles; for example, using fewer awareness supports when a call is not active. Finally, an integrated approach could automatically respect boundaries (e.g., breakout rooms), avoid inconsistent information, and help everyone in a group call establish a shared workspace.

### 3 FORMATIVE OBSERVATIONS

Previous research suggests that tightly coupled remote work is difficult, even with video chat [20]. To build on this understanding in the case of collaborative content, we conducted two formative sessions in which groups of five (7 man, 3 woman; participants

were all office workers employed within a research unit; remote work experience: all were employed remotely at the time of the study) collaborated with existing tools (see Figure 2).

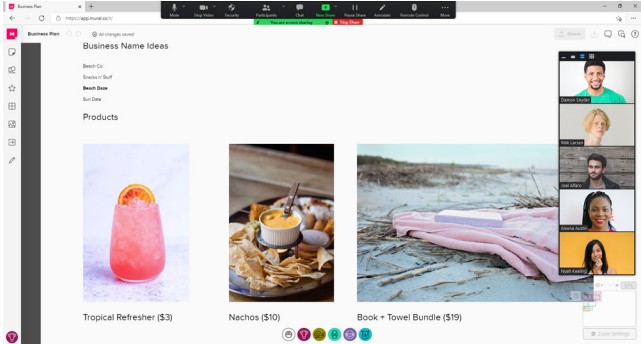

Figure 2:  A collaboration setup used in our sessions: Zoom and Mural (people and content are illustrations, not from our data).

The sessions used Zoom and two real-time collaborative apps: Microsoft Word online and Mural (a digital whiteboard). The task was to create a business plan for two prompts: "A stall on a tropical beach full of tourists" and "A kiosk in a busy mall". Group members were assigned a role—Product Developer (three people) or Writer (two people)—and worked on the following activities: (1) create a name for the business *[All Roles]*, (2) create 10 products, each with a name and image *[Product Developers]*, (3) write a paragraph explaining why people should come to the new business *[Writers]*, and (4) agree on prices for the products *[All Roles]*.

Each group completed the task twice, once using Microsoft Word and once using Mural (the order was switched between groups). For each tool, participants received an overview of available collaborative functionality (Microsoft Word: list of editors, jump to others' cursor; Mural: list of editors, shared selections, telecursors, mini-map, jump to or follow others' locations), then collaborated for 12 minutes. The prompt and roles differed for each tool.

The collected data included participants' screens, audio and video, and questionnaires after each task (NASA-TLX [67] and questions about who they worked with most, what parts of the task they worked on most, and any issues noticed while collaborating). A final questionnaire asked about preference between Word and Mural. Lastly, a semi-structured interview explored the group's organization, feeling of connectedness, and awareness of others.

### 3.1   Observations
We reviewed the recordings and survey data to identify issues participants encountered while collaborating remotely.

**Audio channel limits small-group work.** Our observations suggested that during the middle phase (the role-specific tasks) the product developers tended to occupy the audio channel. Annotating the recorded calls showed that product developers spent a total of 17.73 minutes speaking while writers spent 11.04 minutes speaking (total across both groups and tasks). The cause for this disparity may simply be the larger number of product developers, a reluctance to break into conversation on the part of the smaller subgroup, or the fact that writing work does not facilitate multitasking and discussion. This finding suggests that multiple subgroups may not benefit equally from a shared audio channel.

**Written content can be more difficult to get feedback on.** In one group, a writer asked for others to check over their paragraph, but no one did. In the other group, a writer said they were not happy with their paragraph and others should take a look, but again no one did. In contrast, ideas for products or names, which could be raised verbally, generally received quick feedback from others.

**Misunderstandings and duplicated work were common and often unnoticed.** In several instances, multiple people added the same product or created a heading and area for the same section. In several other cases, recordings showed two people simultaneously searching for images of the same product; this lack of coordination was not revealed until they returned to the workspace to see an image already added. While duplicated work can be desirable in some circumstances (e.g., brainstorming), the duplicated work we observed was silently discarded, not considered as an improvement.

**Collaboration tools infrequently used**. Recordings revealed that participants did not use jump and follow. While recordings cannot show with certainty whether participants looked at Mural's mini-map, none interacted with it, and several participants were unaware of changes outside their viewport (which it displays). The infrequent usage may relate to the session length, task requirements not calling for such interactions, or friction when using these tools.

### 4   DESIGN CONSIDERATIONS FOR THE PEEK-AT-YOU SYSTEM
In addition to our formative observations, we based our design on five team configurations and four design goals.

### 4.1   Team configurations to support
Researchers have developed frameworks for describing mixed-focus collaboration [52]–[54], but not for fully remote synchronous collaboration. Guided by these frameworks and our formative observations, we focus on five key group configurations. To describe these configurations, we use concepts identified in previous work [52], [53] that we define as follows: a team is the collection of individuals collaborating in a call; a subgroup is a unit of two or more people collaborating and a main group is a special subgroup that is maintaining the conversational floor. We introduce the "main group" unit because of poor support for parallel conversations in video calls [68] (e.g., this was observed with the product developer subgroups in our formative observations).

Considering previous work [52], [53] and the peculiarities of video calls (e.g., breakout rooms, limited parallel conversations), we define five key configurations to support (see Table 1):

1. **Individual Work**: each person works alone

2. **Individual + Subgroups Work**: some people work alone, while others work together in pairs or small groups

3. **Subgroups Work**: all people work in pairs or small groups

4. **Splintered Team Work**: most people work together in a main group, while a few work individually

5. **Team Work**: all people are working together in a main group

To define subgroup more precisely than "people collaborating", we considered states identified in co-located work [54] and adapted them to a remote and task-agnostic context by

recognizing two key concepts: sharing a view and discussing. These concepts have been used in coding remote [69] and hybrid [53] collaboration, and help formalize differences between conversational and visual feedback seen in our formative observations. Therefore, we consider four subgroup states: *not existing*, *discussing*, *working on the same content*, and *working on the same content and discussing*.

Table 1. Our five team configurations illustrated for a team of eight.

| | | Individual States | | |
|---|---|---|---|---|
| | | Working individually | Working with a subgroup | Working with the main group |
| Group states | Individual work | 👥👥👥👥 | | |
| | Individual + subgroups work | 👥👥👥 | 👥👥 | |
| | Subgroups work | | 👥👥👥👥 | |
| | Splintered team work | 👥👥 | | 👥👥👥 |
| | Team work | | | 👥👥👥👥👥👥 |

The heart of mixed-focus collaboration is fluid transitions between states [8]. Therefore, it is important to consider not only the configurations or states that individuals, subgroups, and teams can take on, but also the variety of transitions that can occur between them (e.g., from Individual Work to Subgroup Work). The transitions from individual to subgroup or team work involve particular challenges: unlike face-to-face collaboration, physical movements and reconfigurations of the workspace that can support transitions [58] are not possible. People must quickly and accurately understand what others are working on to assess when a transition is appropriate and whether it has succeeded. On a subgroup level, transitions between discussing and not discussing are mainly constrained by the availability of the audio channel. Transitions from not sharing a view to sharing a view can be more difficult to do quickly without system support.

## 4.2 Design goals

Based on existing literature, our configurations to support, and formative observations, we address blockers to remote mixed-focus content creation with a system designed around four goals:

**DG1. Build awareness when and where needed.** Systems for mixed-focus collaboration should actively build awareness of collaborators' actions and positions, rather than relying on passive indicators. Previous approaches to this goal include detecting references to documents in conversation to surface relevant files [70], [71], detecting periods of inattention and using highlighting, motion traces, or replays to catch up [72], or manually configuring avatars that can notify users of certain actions by others [73]; we focus on automatically and continuously supporting awareness.

**DG2. Support understanding of conversation.** Conversations can be difficult to understand when views differ between collaborators, and existing solutions for maintaining awareness are passive in these situations. Previous approaches to this goal include awareness widgets like mini-maps [7], [43], detecting content references in text messages and determining the probability of misunderstandings based on gaze detection [74], or sharing gaze positions with other users [33, p.]. We focus on using integration with the interpersonal space to go beyond traditional awareness indicators without requiring gaze detection hardware.

**DG3. Allow fast and simple transitions between collaborative states.** Understanding what others can see and

establishing shared views should be quick and easy, supporting transitions. We focus on supporting lightweight transitions that occur without changing the communication medium or work artifact. Previous approaches have also enforced additional structure. One structured approach is turn-taking of control (e.g., driver and viewer roles for editing documents [75] or music live coding [76]); however, research suggests that verbal and non-verbal communication can obviate the need for rigid turn-taking protocols [77] so we focus on more flexible state transitions in our video-chat based system. A second structured approach is handoff of information (e.g., using a specialized visualization for collaborative sensemaking [78]); however, for collaborative content creation, we focus on leveraging views and positions in the existing artifact to support transitions.

**DG4. Means for lightweight feedback.** Assistance and feedback should be easy to provide. Visual communication should be supported for gestures, referencing, and times when the audio channel is occupied. The primary previous approaches for lightweight feedback include telecursors [40] and screensharing [38]. We focus on simpler and more transient view-sharing and enabling gesture-friendly cursors on these views.

## 5 PEEK-AT-YOU: NEW COLLABORATIVE SYSTEM LEVERAGING INTEGRATION

Based on our design goals, we created Peek-at-you, a set of collaborative features developed with our four design goals in mind, implemented as a Chrome extension that extends Google Docs. The design reflects the specific case of document editing, which is the focus of our evaluation, but the features are designed to apply to various artifacts (e.g., slides, digital whiteboards, 3D models, or interface designs). Our system focuses on allowing a tightly coupled group to successfully leverage the rich communication and awareness possible within a single video call; therefore, we did not include other established methods of managing communication that split up conversations (e.g., text chat [79], breakout rooms, or spatial video chat [42], [80]–[82])

### 5.1 Conversation-based position indicators

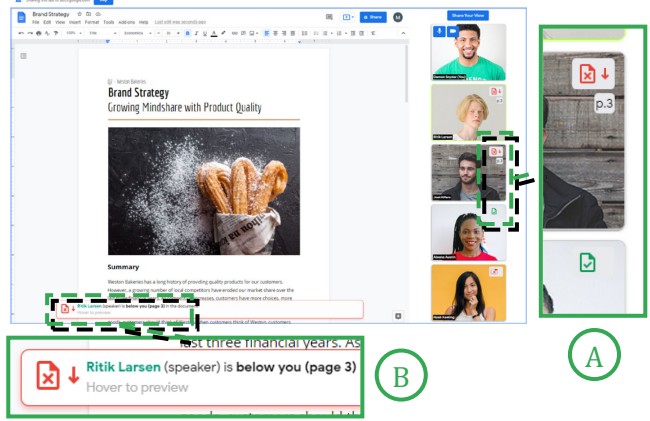

Figure 3: Video call integrated into Google Docs using Peek-at-You. Conversation-Based Position Indicators appear on others' video feeds (A) and at the bottom of the work area (B).

Position indicators help collaborators understand where in the document others are working (DG2). Because our system integrates video chat with collaborative software, this information can be surfaced where and when we expected it to be most useful (DG1). First, icons are shown in the corner of each users' video feed (see Figure 3, A), indicating the collaborator's current page

and whether they are in the same place, above/below, or in another tab. This is based on our observation that users who are speaking might erroneously assume they are looking at the same thing. Clicking the icon scrolls to the collaborator's position (DG3). Placing awareness supports on others' video feeds is unique approach that does not use screen space within the task area. Second, an active speaker popup is shown at the bottom of the work area when a collaborator is speaking (see Figure 3, B), containing the same icons as in the speaker's video feed and a description of their state (e.g., "below you (page 6) in the document" or "in another tab"). For users focused on the shared workspace, this actively shows relevant information from the interpersonal space without the need to scan the indicators in the video chat. Conversational position indicators on video feeds provide many components of awareness: presence and identity via video feeds, location and view via position indicators, and action and artifact via jumping. Further, understanding others' viewpoints and navigating to them afford the fundamentals of we-awareness [49], a key requirement for discussing content. This is important as effective communication and organization could allow more content-focused work time [37].

## 5.2 Speaker's view peeking

Quick and fluid transitions between individual and shared work are key to mixed-focus collaboration [8], and getting feedback is an important component of creating content together [37]. Therefore, the active speaker pop-up described above can be hovered to quickly preview the current speaker's view (see Figure 4). This functionality is inspired by our observation that content such as writing can be difficult to get feedback on, and people may be hesitant to leave their position to see what someone else is talking about (DG2, DG3). When peeking someone else's view, viewers can react using a set of five reactions (DG4): thumbs up/down ( 👍 / 👎 ), eyes ( 👀 ), ok ( 👌 ), and thinking ( 💬 ). Viewers can also use cursor trails (colored dots that temporarily appear as their mouse cursors move on the preview); these specialized telecursors are well suited to gesturing [83] (DG2, DG4).

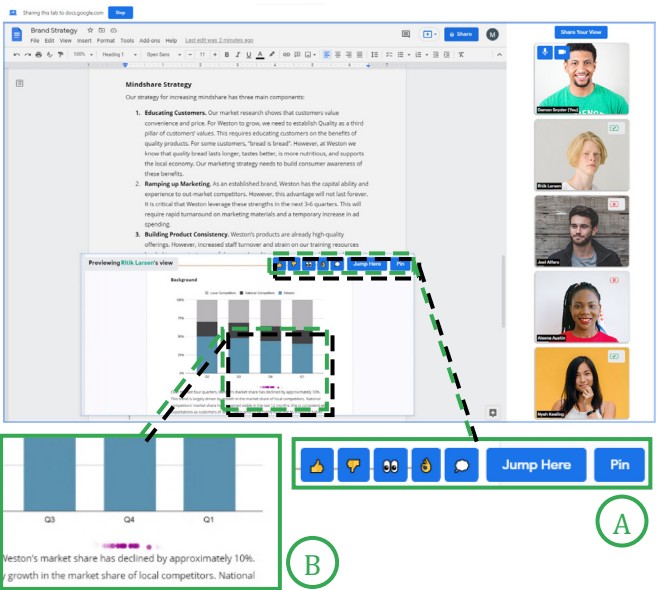

Figure 4: Peeking a collaborator's view using our system. Viewers can react (A) and gesture with cursor trails (B)

## 5.3 View pushing

In addition to peeking others' views, the system allows participants to quickly share their own view with everyone by clicking a "share view" button (DG3, see Figure 5). Like view peeking, this feature is based on the need to easily get feedback and transition between working styles; this feature particularly supports transitions to full-group work. Other collaborators can dismiss the shared view if it is not relevant to them, and the sharer sees a list of current viewers (DG1). Viewers can use the same reactions and cursor trails that are available when peeking (DG4). For quick transitions, pushing ends any existing view push (DG3). By offering both View Peeking and View Pushing, the system provides robust support for passive and active maintenance of the action, artifact, and view components of workspace awareness. Additionally, by enabling a shared context for conversation, these features may allow collaborators to discuss more nascent aspects of workspace awareness such as intention.

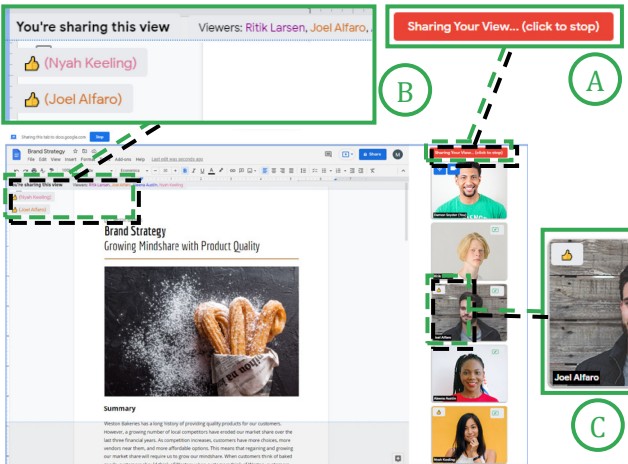

Figure 5: Pushing a view to collaborators. Pushing is started or stopped with one click (A), lists the current viewers (B), and shows reactions on the content area (B) and video feeds (C).

## 5.4 Prototype implementation details

Our prototype extends an existing groupware system, Google Docs. Although our formative observations were done using Microsoft Word, we chose Google Docs, as its HTML was easier to extend. We integrated video chat into the Google Docs page via a sidebar on the right side, where the active speaker is highlighted with a green outline, and added the previously described features.

The prototype extends Google Docs using a Google Chrome extension. React and Typescript are used to inject the system's interface, capture camera and tab feeds, and track viewports. The video chat uses WebRTC, with Kurento Media Server [84] for server-side recording and hark.js (https://github.com/otalk/hark) for active speaker detection. A NodeJS server and WebSockets are used to sync collaborators' states (position, active tab, view shares, etc.). Position icons show whether collaborators' scroll positions are the same as the user (>66% viewport overlap) or above/below, and whether others are sharing views or in another tab. Users share the tab when joining; this stream is transmitted for recording and forwarded as needed for view sharing.

## 6 SYSTEM EVALUATION

We studied six groups of five people in a mixed-focus content creation task to gather initial feedback about our Peek-at-You system. The study was approved by the institutional ethics board.

### 6.1 Task

The study task involved creating a plan for a hypothetical business merger. It differs from the business plan task used for our formative observations in two ways. First, the task begins with an existing document, ensuring sufficient content for the need of positional awareness. Second, an editor role was added, to ensure to increase the working configurations and transitions. Participants were assigned one of three roles: Writer (two participants), Marketer (two participants), or Editor (one participant). Groups received a document containing information about a fictional company and three candidate companies for the merger. Their task was to select the best candidate and plan for the new company:

- (All Roles) Review background info on the company and three merger options, then choose a merger option
- (Writers) Write one or two paragraphs for investors about why the merger will help the company grow
- (Marketers) Create a New Company Name, New Hero Offering, and Marketing Plan for the company
- (Editor) Help out the others as needed and check all new content for quality/consistency

The evaluation included two conditions (Video Chat Only and Peek-At-You) in a within-groups design, so two merger planning documents were created, allowing groups to perform the task twice. The first document described a bakery chain choosing between dessert bakery, deli, and smoothie chains. The second document described a sportswear retail chain choosing between local-focused, sporting equipment, and yoga clothing chains. Each document included background information, a product summary, and a SWOT analysis for the company (1.5 pg.); investor statement placeholder (0.5 pg.); overview, strengths, and weaknesses for each merger option (3 pg.); merger decision placeholder (1 pg.); new name and hero offering placeholders (1 pg.); and marketing plan placeholder (1 pg.). Placeholders included a reminder of what to add and scratch space for drafting.

### 6.2 Procedure

The study was conducted remotely via Zoom, except for the collaborative work which used the video chat integrated in our prototype system, lasting 75 minutes. After giving informed consent, and completed the task twice, once with only the video chat only mode (Video Chat Only) and once with all features enabled (Peek-At-You). Each time, they were given instructions and roles to collaborate using Google Docs. In the Video Chat Only condition, the instructions were pointed to collaborative functionality of Google Docs such as the editor list at the top of the page. In the Peek-At-You condition, the instructions were a brief interactive tutorial. Next, participants opened a copy of the task instructions in a separate tab for reference and spent 15 minutes collaborating. After finishing, participants also completed a survey about the experience. The order of conditions and roles was counterbalanced between groups and tasks.

### 6.3 Measures

After each condition, a questionnaire was used regarding:
- NASA-TLX [67]
- Collaborative Experience: three items rating participants' Distractedness ("I was frequently distracted as I tried to work"), Awareness ("I had a good sense of what other people were working on at all times"), and Understanding of Discussion ("It was easy to follow the ongoing discussion"). on a 7-point Likert scale (strongly disagree to strongly agree).
- Feedback: open feedback about the system or experience.

After completing both conditions, a final survey asked which condition participants preferred and why. Finally, a brief (10 minute) semi-structured group interview was conducted regarding ability to get feedback from others, ability to understand others, desire and ability to maintain awareness, and reasons for using or not using Peek-At-You's features. Participants' screens and video calls were recorded during the tasks and log data was collected.

### 6.4 Participants

Participants were recruited from within our institution using email and Slack channels and compensated through an internal award program (approximate value pre-tax 70 USD). Twenty-nine participants were recruited in six groups of five people. Due to a last-minute cancellation, one group completed the study with four members rather than five; this was accommodated by omitting the editor role. In total, 29 participants completed the study (17 Female, 12 Male; Age: mean=29.4, SD=9.3). Their professions were varied (Software Developer/Engineer=11, Design/UX=3, Analyst=5, Researcher=3, Community Support=2, Management/Supervision=2, Marketing=2, Legal=1), but all were experienced with remote work. Most participants were unacquainted, but none were coworkers. While our sample size and the complex dynamics within a five-person group interaction did not allow us to account for these instances within our analysis, we expect the fixed task, randomly assigned roles, and within-group study design minimized any potential effects of these differences and we did not make observations related to acquaintedness.

### 6.5 Evaluation findings

Our findings focus on understanding how much participants used Peek-At-You's features, their preferences for our system or the Video Chat Only condition, and their feedback on each condition.

#### 6.5.1 System usage

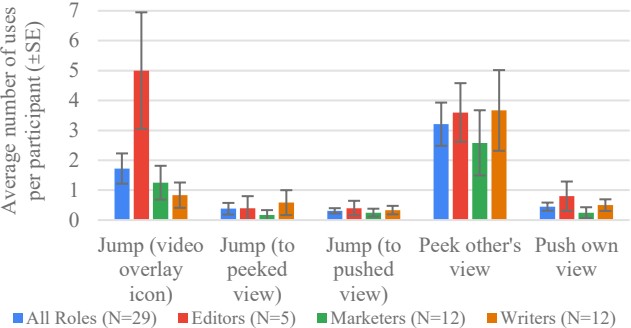

Figure 6: Collaborative feature usage per-participant. Averages are provided across all participants and per-role.

Usage of our system's features was analyzed using log data. In the 15-minute session, participants used the jump functionality of the video overlay icons an average of 1.72 times, the peek functionality an average of 3.21 times, and the push functionality an average of 0.45 times (see Figure 6 for details). While push was used less than peek on a per-participant basis, it is worth noting that pushes affect the entire group whereas peeks are displayed only to the local user. Another important caveat to these usage numbers is that they do not capture how often participants looked at the Conversation-Based Position Indicators, usage which is better captured through survey and interview responses.

#### 6.5.2 Survey responses

Participants' responses to the NASA-TLX were similar in the Video Chat Only and Peek-At-You conditions (see Table 2).

Table 2.   NASA-TLX responses. Values are mean (SD).

|  | Video Chat Only | Peek-At-You | Wilcoxon Signed-Ranks |
|---|---|---|---|
| **Mental Demand** | 6.48 (2.11) | 6.48 (1.45) | Z=-0.06; p=.95 |
| **Physical Demand** | 2.48 (2.52) | 2.03 (2.18) | Z=-1.47; p=.14 |
| **Temporal Demand** | 6.07 (2.12) | 6.52 (2.13) | Z=-0.76; p=.45 |
| **Performance** | 5.24 (2.71) | 4.59 (2.21) | Z=-0.85; p=.40 |
| **Effort** | 5.93 (1.98) | 5.76 (2.08) | Z=-0.17; p=.87 |
| **Frustration** | 4.86 (2.67) | 4.69 (2.04) | Z=-0.38; p=.70 |

Responses to the Collaborative Experience questions show some differences between the Video Chat Only and Peek-At-You conditions (see Figure 7). These were tested using Wilcoxon Signed-rank tests. Participants expressed greater agreement regarding their understanding of the conversation in the Peek-At-You condition, but this difference was not significant (Z=-1.101; p=0.267). Participants rated their awareness of collaborators higher in the Peek-At-You condition (median=Somewhat agree) than in the Video Chat only condition (median=Somewhat disagree); the difference was significant (Z=-2.15; p=0.03). Participants did not rate their level of distraction significantly differently in the two conditions (Z=-0.26; p=0.80).

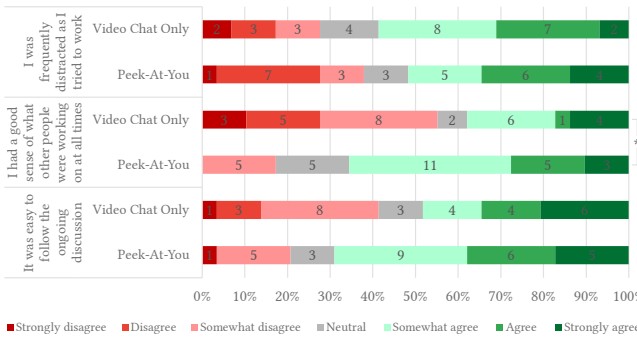

Figure 7:  Responses regarding Collaborative Experience (*p<.05).

A majority of participants preferred the Peek-At-You condition (n=21). A subset of participants preferred the Video Chat Only condition (n=8). Among participants preferring Video Chat Only, roles in the Video Chat Only condition were Editor (N=2), Writer (N=4), and Marketer (N=2) while their roles in the Peet-At-You condition were Writer (N=3) and Marketer (N=5). Participants provided open-ended feedback regarding the reasoning for their preferences. Among the participants who preferred the Video Chat Only condition, four did not feel the new features were needed to maintain awareness, or felt that a high degree of awareness was not needed in this task. The other four found the features distracting due to rapid visual changes. Five of these eight participants experienced Peek-At-You with the Marketer role; while the sample is not large enough to test for significant, it is possible that the marketing role in particular was well suited to verbal discussion required less in-artifact coordination. Among the participants who preferred the Peek-At-You condition, reasons for the preference were varied but related to usefulness in supporting awareness and understanding. A qualitative analysis of participants' feedback was performed to provide greater insight into these perceptions.

### 6.5.3   Participant feedback

To analyze participants' experiences and feedback we used an open coding approach, where two authors separately coded transcripts until no new codes appeared, then reviewed each other's coding for agreement; this included data from two groups.

The first author coded the remaining data and identified eight themes, which were merged into six themes after discussion between two authors.

**Peek-At-You aids awareness**. Participants found the Peek-At-You system to be interactive and helpful in maintaining awareness of collaborators' locations, roles, thought processes, and task progress. The conversation-based position indicators helped participants stay aware of others' locations and focus on what they wanted to share. Displaying collaborators' positions helped communicate roles by being able to see which areas of a document everyone is working on. Tracking indicators over time can also reveal thought processes such as referencing one part of the document to help with writing elsewhere; P11 explained the system "definitely helped us understand, like, who was working on what and what they were, what their thought process was." In addition to process, position indicators can communicate progress on a task: "even, I think, something as simple as whether we've finished reading, and that was easy to understand in the [Peek-At-You condition]" (P26).

**Peek-At-You supports conversational understanding.** Participants found it was easier to understand what others were speaking about with our system, with P13 stating that "it was easier to know what someone else was talking about or referring to". This suggests that the additional awareness of others' positions, thought processes, task progress, and roles provides context that makes following the conversation easier.

Participants specifically appreciated the popup showing the active speaker's location, as it aided with following the conversation. For P26, "it was really helpful to see the speaker's view and be notified when I was not on their view." P11 found the popup "was a little bit distracting sometimes, but it definitely was helpful." This suggests both roles of the popup (i.e., warning when the listener is not seeing the same part of the document and view peeking) are valuable for conversational understanding.

**Peek-At-You aids transitions.** Participants reported that Peek-At-You's features were helpful for transitioning to mixed-focused collaboration. For example, P21 found it "easier to track others, share progress, find one another". Position icons aided in grouping up; in one instance P25 explained "I couldn't find the section where we were supposed to be writing and I was able to jump up to where P27 was, was taking a look. So yeah, I found it helpful." Jumping to others' video feeds also helped with temporarily transitions, e.g., "I was doing the marketing stuff, so I was like, looking at what, P10 and P11 were like adding just so that I could like, know what was happening like on the other part, and yeah I was just like jumping to their pages with the little, little icon on the video." More generally, P23 found "the features allowed me to quickly hop back and forth between where other people were looking and working."

Pushing a view was a quick way to ensure everyone was looking at the same thing. As P4 explained, "I was able to share my screen on the merger page and everyone else could pop-up on my screen, so they didn't have to scroll all the way back up." P17 noted that Peek-At-You's features "made it easier to share views and get input without having to completely leave the work you were doing." In contrast, P6, P8, and P11 described challenges to transition to group work in the Video Chat Only condition.

**Audio channel aids awareness but difficult to share.** Some participants mentioned that the audio channel helps maintain awareness. However, many participants reported that sharing the audio channel was challenging. For example, at times "others had

to take a pause until the main conversation was over or find another way to speak without disruption" (*P3*). This was especially apparent when participants were working in small-group configurations. When *P1* and *P2* worked together, *P1* found "it was hard to coordinate with *P2* because we didn't want to, like, talk over like *P3* and *P4* talking." *P16* likewise found that "it was annoying trying to have a discussion with just part of the team while other[s] were, are having a conversation."

While breakout rooms or selective muting are possible solutions, these approaches are also likely to reduce awareness within a group. Collaborative features like the ones in our system may attenuate the need for breakout rooms by reducing verbal articulation work (the work of working together [85], [86]).

**Collaborative features can be distracting.** Although helpful with awareness and understanding, some participants found certain aspects of our system distracting, such as rapid visual changes and shared views taking up too much screen space. To manage these distractions, participants suggested using collaborative features only during certain phases or being able to turn them on and off as needed. For example, one participant felt the Peek-At-You feature was only important initially, during brainstorming and discussion, while another suggested having the feature be toggle-able.

**Collaborative features may be more useful with experience or in other tasks.** Participants also explained that because the features were new, they may not have fully learned or thought to use all of them during the study. *P8* explained that "since the UI was new, we were getting distracted because of that", but "the more we use this tool, the more efficient ways we will find to make the most of it." *P23* felt similarly: they "didn't use some of the features consciously due to familiarity. With more exposure to the extension and conscious effort it will become more natural."

Participants suggested system would be useful in other scenarios, such as collaborative work or presentations with multiple slides, as it would reduce the amount of scrolling (*P6*). They highlighted the usefulness of the sharing through push and peek, as well as the preview feature for keeping track of others' progress without interrupting their own work.

## 7    DISCUSSION

We discuss how our system supports fluid working configurations, why existing applications should enable extensibility to support the functionality of Peek-At-You, and adapting our system to reduce distractions.

### 7.1    How in-the-moment indicators support transitions

Our evaluation shows that Peek-At-You supports smooth transitions in mixed-focus collaboration by increasing awareness of co-editors' positions, roles, thought processes, and task progress. Survey data confirmed that our system supports this type of awareness, which is important for identifying opportune times to interrupt the current working state of the group during transitions and to understand when transitions into subgroups or a main group succeed. For example, transitioning from individual to subgroup work may involve identifying others with the same role. Similarly, transitioning to teamwork may involve identifying when everyone has made sufficient progress on their individual work first. More generally, transitions are aided by understanding the processes of collaborators and choosing an opportune time to interrupt the current working state of the group [6].

Sharing views can also support transitions. While sharing or viewing of private information is simple when face-to-face [87],

we show that one-click and conversational interactions can make view sharing equally easy in a remote context. View pushing and peeking further allow to jump to the location for a complete editing experience. Unlike spatial video chat systems that allow users to share views via screensharing and move participant videos around on top of the shared view to group up around a particular element [42], our system supports full content control after jumping.

Using awareness of others' positions and actions is a quick and lightweight way to transition into different working configurations while maintaining awareness of the rest of the group. However, traditional approaches, such as breakout rooms or position-based audio muting [80]–[82], provide stronger separations between groups. While enabling focused work, this limits awareness of other subgroups, leading to challenges, such as unawareness about what a breakout group is working on or when to interrupt.

### 7.2    Peek-at-you vs. our formative observations

We return to the four themes identified in our formative observations to compare the findings to our evaluation study.

**"Audio channel limits small-group work."** Our system's awareness features reduced the need for verbal articulation work, which may ease the experience of sharing an audio channel. However, sharing an audio channel was still difficult at times, and other solutions such as selective muting, subgrouping, or breakout rooms, are needed scale to arbitrary group sizes.

**"Written content can be more difficult to get feedback on".** Participants found view pushing and peeking useful to quickly establish a shared view. Grouping up around a shared view is an effective way to gather feedback on writing, as it does not require reading the text aloud or losing one's position in the document.

**"Misunderstandings and duplicated work were common and often unnoticed".** Participants noted that our system supported conversational understanding, with position indicators being helpful for tracking discussions. While we could not make direct comparisons with our formative observations, participants indicated that position indicators aided them in assessing what others were working on, helping to avoid duplications.

**"Collaboration tools infrequently used".** Participants used Peek-at-you over 5 times on average, which compares favorably to the formative observations, where collaboration tools (e.g., jump/ follow) were not used. It's worth noting that the longer content in the evaluation task makes a direct comparison difficult. However, placing collaboration tools on video feeds may have also made them easier to access, contributing to increased usage.

### 7.3    Managing distractions

Mixed-focus collaboration involves processing a lot of information, including video/audio communications, real-time artifact changes, and awareness widgets. Our system supplies real-time information, which some participants found distracting due to rapidly changing icons or overlays taking up screen space. However, our questionnaire did not show an overall increase in distractedness when using Peek-at-You, possibly due to distractions inherent to real-time collaboration overshadowing distractions related to our system. Alternatively, increased distractions from the system may have been balanced by a decrease in other distractions, such as improved conversational articulation or better leverage of interruption strategies [45].

Though a degree of distraction is inherent to mixed-focus collaboration, the design of collaborative systems involves

tradeoffs between maintaining awareness and avoiding distractions [7]; the desired balance may depend on many factors including group size, task, artifact type, and roles. Because some participants in our evaluation cited distraction as a drawback, we suggest four design iterations that could reduce distractions. First, position indicators could use "calm design" [88] by displaying only a binary red/green status light until hovered and varying the active speaker notification [89] based on speaking and working activity (Figure 8, left). Second, shared views could be sized more precisely to manage screen space. Currently, our prototype sizes shared views based on the window aspect ratio of the sharer and viewer, but this may result in a larger than intended preview in some cases. Third, view pushing could incorporate a "consent" mechanism where shared views are small but expand if hovered (Figure 8, right). This approach may offer some of the benefits of continuous gestures like moving and resizing elements in spatial video chat systems [32], [42], [80]–[82], while still being compatible with a standard scrolling document interface. Fourth, a focus mode could be added, which would hide collaborative features, selectively present audio using roles or proximity, or even hide others' edits. Video overlay icons could signal which collaborators are in focus mode. This may also make the system more inclusive (multiple participants cited ADHD as a particular motivator for minimizing distractions) and support hybrid work that includes loosely coupled phases [51].

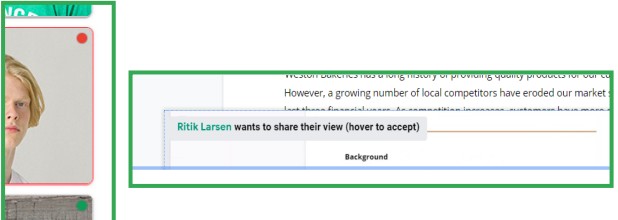

Figure 8:  Potential design iterations: (left) a calm design that uses binary status lights instead of icons and a color-coded outline instead of the active speaker popup; (right) a pushed view that uses a consent mechanism before appearing full size.

### 7.4    Comparing methods of sharing views

For peeking / pushing views, we implemented view sharing through video streaming of the user's view, with the option to navigate to their view by clicking the position icon. We relied on a video stream because the tab video was already being streamed for recording and deep integration is difficult with a closed-source application (Google Docs). However, using local rendering for view sharing in collaborative software would provide several benefits, such as bandwidth and quality improvements, increased accessibility, and making the multiple views editable. Regardless of rendering approach, integrating shared views can preserve privacy compared to general-purpose screen sharing, as it only shares content that collaborators already have access to [23].

Jumping to someone's view offers an alternative to temporary view sharing, but it can cause context loss for the person jumping. A possible solution is to blend jumping and peeking, as Gutwin et al. [43] did by holding the right mouse button to jump to a collaborator's view and releasing it to jump back. A "back" button could also be shown to aid within-document navigation.

### 7.5    Supporting integration of group calls and collaborative apps

Currently commercial apps are replacing traditional screensharing with embedded collaborative apps in group calls. For example,

Google Docs now integrates video calls and Zoom allows third-party apps to integrate with the shared stage. To enable consistency between collaboration and communication apps, we argue that APIs for UI extensibility and data access are needed—e.g., for assigning an icon to be displayed on top of a participant's video feed or receiving notifications about the current speaker.

Because the Peek-At-You system requires sharing data between the collaborative app and communication app, users can only benefit from such features using specific tools that support this integration. However, if communication platforms establish standard ways of sharing such data, this limitation may be minimized as more apps in a team's workflow support similar integrations. These APIs would also be useful in several other ways (e.g., selecting video feeds to show based on viewport proximity, linking call recordings to artifact edit histories, or displaying icons on top of video feeds to help people understand others' emotions).

### 7.6    Generalizing to other tasks, groups, and artifacts

We designed with relaxed-WYSIWIS systems in mind, but focused on content creation using a document editor for prototyping and evaluation. Different types of systems would require adaptations to represent positional indicators. For example, a digital whiteboard with 2D navigation may need to represent "up, left, and zoomed out", while a presentation or interface design application may need to represent that a collaborator is on a different slide or screen. 3D applications may present even more challenges, but could leverage arrows [90] or a Viewcube [91]. Different types of systems would require adaptations to represent positional indicators.

Relaxed-WYSIWIS groupware may allow users to have different object formatting and representations [40], which makes establishing a shared view challenging for two reasons: "jumping" to another person's view incurs a significant loss of context and determining whether people are currently sharing a view may be difficult (e.g., if two people see the same table in a spreadsheet but have applied different data filters). Our view peeking and pushing features preserve context  and guarantee identical object representation, which may be particularly helpful in these contexts.

Our system's collaborative features were designed to support a variety of tasks within content creation process as part of an individual (reading), a team-level (choosing a merger target), and small-group activities (generating investor statements and marketing materials). While other tasks may require a different configurations, our design does not impose a specific ordering or structure for collaboration; therefore, while not yet tested, our system may by useful for other mixed-focus collaboration tasks such as brainstorming, decision making, or reviewing.

Our system could scale to larger groups, but stricter approaches for supporting subgroups may be needed (e.g., breakout rooms or audio filtering based on spatial positioning [80]–[82]). The integration of communication and collaboration leveraged by Peek-At-You could be helpful in these cases, such as using collaborators' proximity within a document or other artifact to select which video feeds or audio feeds to present, providing the most relevant awareness information.

### 8    LIMITATIONS & FUTURE WORK

The proposed system in this work is tailored to a specific context and may require adaptations for other contexts. While our

experimental setup allowed us to recruit groups of a non-trivial size (29 participants in six groups), include a Video Chat Only condition, and recruit participants familiar with remote work, studying a single group size, task, and artifact type limits our ability to draw strong conclusions about the generalizability of our system. Future research should test the system in various contexts to evaluate its generalizability and effectiveness. Additionally, longer-term deployments of the system can help to understand how it can support sustained collaboration over time. In particular, testing the system's effectiveness in projects involving multiple collaborative tools that do not all support the integration required for Peek-At-You will be a key step to ensuring generalizability. Future work should also consider how experience affects system usage, as some participants found that our study's duration limited their ability to learn and leverage all the features. Finally, future work should further study how integrated tools can support hybrid asynchronous-synchronous collaboration.

## 9 CONCLUSION

In summary, we contribute to research in mixed-focus content creation in multiple ways. First, we build on existing understandings of mixed-focus collaboration and our formative observations of fully-remote collaboration. Second, we then design Peek-At-You, a system of collaborative features that leverage understanding of conversation and collaborative actions to increase awareness, facilitate understanding, and support the transitions needed in mixed-focus collaboration. Finally, we evaluate the system in groups of five collaborators, demonstrating that it can foster the knowledge and actions we intended to support. By enhancing remote collaboration, we contribute to making benefits of collaboration available for remote content creation.

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
