# OpenReview forum: "Peek-At-You: An Awareness, Navigation, and View Sharing System for Remote Collaborative Content Creation"
_graphicsinterface.org/Graphics_Interface/2023/Conference_SD — GI 2023 - second deadline_

### Official Review · Reviewer_DefQ · 2023-04-27
**Well motivated system, in need of further iteration.**

**Rating:** 5
**Confidence:** 4

**Review:**

This paper introduces a system to support mixed-focus collaboration. A prototype system integrates a zoom-like interface with google docs and includes a set of features for sharing views or peeking at the position of others, with the aim of increasing awareness during ad-hoc collaborative practices. The prototype is evaluated with 6 groups of users.

This system is interesting and I can understand the motivation for integrating these features in a single system. This comes with the drawback however, that other tools can not be integrated into the workflow. This is a substantial limitation and should be addressed in the discussion.

The paper feels long relative to the contribution and I believe the contributions could be expressed more succinctly. I didn't see a strong value in the formative study. This also didn't include commonly used features like breakout rooms. I suggest to either remove this study or more clearly link the outcomes to the design goals.

Overall, the system seems promising, but appears to need further iteration to develop the features, as indicated by low adoption in the study. the study results will provide useful formative feedback for further development of the prototype features, as discussed in the paper. In addition to further iteration, possibly including further formative studies, a longitudinal evaluation would help to provide a better picture of the usefulness of the proposed features.

---

### Official Review · Reviewer_SzuY · 2023-05-01
**timely and relevant system/artifact contribution to remote collaboration**

**Rating:** 7
**Confidence:** 4

**Review:**

The paper presets the design of a collaborative system that integrate features of interpersonal communication and shared workspaces for remote collaborative work. The system is specifically designed to support collaborative content creation in which individuals may switch their focus between individual and team work. The graphic interface design and visual features are proposed to support awareness, understanding of conversational content and transitions between collaborative states. The system was evaluated in a within-subject study with 6 groups of users.

Overall, the paper is well-motivated and well-written. It presents a clear and well-structured literature and rationale behind the design goals. The system proposed is also clearly described and presented. The key features are reasonable and appear to well-used by the participants during the evaluation. While these GUI features don't appear to be very novel, they're nevertheless new and useful when they're integrated and applied in the context of online content creation.

The  system evaluation reads more like a pilot study that involves only 6 group of users, and it's done in a short within-subject study. The study is still working well, showing the value of the proposed design. However, the ecological validity of the study is thus limited (which has been acknowledged in the limitation section of the paper).

In summary, the paper would be a great addition to the GI conference by contributing a communicative and collaborative interface that are shown to be useful and practical to remote work.

---

### Official Review · Reviewer_xj5A · 2023-05-02
**n/a**

**Rating:** 8
**Confidence:** 4

**Review:**

The submission discusses a formative observation, the design of a system, and an evaluation of a system for flexible collaboration in teams.

+ Although collaboration, groupware and remote group are a crowded spaces, the more subtle hybrid work addressed here is important and much less considered
+ The research is timely because of the increase of remote collaborative work due to the pandemic and the current popularity of tools that are starting to support both workspace sharing and visual/audio communications (e.g., Miro)
+ The submission contains a full research cycle including formative study, design of new system, and evaluation thereof.
+ The design is well justified based on the formative study and a well grounded literature research (through design principles)
+ Although the features of the tool are not earth-shatteringly original (they are mostly common sense), this is what is sometimes required from interfaces. The design is meaningful and covers sufficiently the set of needs established a priori.
+ Both studies are competent, and their limitations explicitly brought up (e.g. the issue with time of use and familiarity of task and interface).

Overall, I think this is submission offers a solid contribution in an area where it is needed and, simultaneously, where it is difficult to contribute due to the large number of existing systems.